# Persuasive Messages Will Not Increase COVID-19 Vaccine Acceptance: Evidence from a Nationwide Online Experiment

**DOI:** 10.3390/vaccines9101113

**Published:** 2021-09-30

**Authors:** Raman Kachurka, Michał Krawczyk, Joanna Rachubik

**Affiliations:** Faculty of Economic Sciences, University of Warsaw, 00-241 Warsaw, Poland; r.kachurka@gmail.com (R.K.); j.rachubik@uw.edu.pl (J.R.)

**Keywords:** COVID-19, vaccine refusal, vaccine hesitancy, vaccine uptake, vaccine acceptance, persuasion

## Abstract

Although mass vaccination is the best way out of the pandemic, the share of skeptics is substantial in most countries. Social campaigns can emphasize the many arguments that potentially increase acceptance for vaccines: e.g., that they have been developed, tested, and recommended by doctors and scientists; and that they are safe, effective, and in demand. We verified the effectiveness of such messages in an online experiment conducted in February and March 2021 with a sample of almost six thousand adult Poles, which was nationally representative in terms of key demographic variables. We presented respondents with different sets of information about vaccinating against COVID-19. After reading the information bundle, they indicated whether they would be willing to be vaccinated. We also asked them to justify their answers and indicate who or what might change their opinion. Finally, we elicited a number of individual characteristics and opinions. We found that nearly 45% of the respondents were unwilling to be vaccinated, and none of the popular messages we used was effective in reducing this hesitancy. We also observed a number of significant correlates of vaccination attitudes, with men, older, wealthier, and non-religious individuals, those with higher education, and those trusting science rather than COVID-19 conspiracy theories being more willing to be vaccinated. We discuss important consequences for campaigns aimed at reducing COVID-19 vaccine hesitancy.

## 1. Introduction

In many countries, a significant percentage of the population opts out of vaccinations, which leads to serious health risks. Reluctance to vaccinate has become striking during the COVID-19 pandemic, with a significant number of people refusing to take a shot protecting against the virus.

Social campaigns can emphasize various arguments that potentially raise vaccination acceptance. For example, they may portray them as safe, effective, developed, tested, and recommended by doctors and scientists, free of charge, voluntary, demanded by others, and available in limited numbers (therefore psychologically more valuable). Vaccine “passports” may also promise greater freedom to travel.

The effectiveness of these messages is not measured systematically and precisely enough. For example, the increase in vaccination acceptance rates after a social advertising campaign may be the result of other circumstances. Moreover, it is not known which aspect of the campaign was particularly effective. Our study seeks to close this gap using a randomized controlled trial with between-subject manipulation of pro-vax persuasive messages addressing the dimensions previously mentioned.

We conduct our study in Poland; while the country is experiencing extreme excess mortality during the pandemic [1], anti-vaccination attitudes have been on the rise in recent years [1,2], and opposition against COVID-19 vaccines is strong, compared with most other countries [3,4].

## 2. Literature Review

Extensive literature has studied the numerous factors which may affect vaccine acceptance. A comprehensive review can be found in Betsch et al. [5,6], who categorize them using the “five Cs”: confidence in the vaccine, convenience to obtain it, calculation of pros and cons based on available information, complacency (triggered by the assessment that the disease is not very dangerous), and collective responsibility (willingness to protect others). We predominantly focus on the confidence factors, which are easy to address with a social campaign or an online experiment such as ours but hard to influence in vaccine-resistant individuals. 

Studies investigating the determinants of COVID-19 vaccine attitudes (see [7,8] for reviews) typically found them to be more negative among low-income [9,10,11], less educated [3,12] populaces, and among ethnic minorities [11,12] and the young [3,11]. Some authors [9,10,11] identify higher COVID-19 vaccine acceptance in males but a major study by Lazarus et al. [3] shows the opposite. International comparisons suggest the most positive attitudes are in Asian countries with a high level of trust in the central government and more negative in Central and Eastern Europe [3,4], possibly as a legacy of Soviet communism [13]. 

In Poland, we are aware of surveys ordered by newspapers, which typically confirm the aforementioned demographic effects, namely higher COVID-19 vaccine acceptance in older individuals, males, people living in big cities [14], and those with higher education [15]. Very recently, Sowa et al. [16] reported a study confirming these results and additionally emphasizing the role of belief in conspiracy theories and views about vaccine side effects.

All of these reported patterns are correlations, thus we cannot establish any causal links. We are aware of but a few experimental studies addressing hesitancy to vaccinate against COVID-19. In Palm et al. [17], a convenience sample of US-based Amazon MTurk workers was targeted in August 2020. Compared with the control group receiving no additional information, those who received a message about the safety and efficacy of the vaccine were more likely to say they would take it (and, of less practical importance, some manipulations decreased the willingness to be vaccinated). 

In perhaps the most comprehensive study to date, conducted in June and July 2020, Schwarzinger et al. [18] surveyed a representative sample of French citizens aged 18 to 64. The authors implemented a discrete choice experiment approach, presenting the respondents with a series of eight choice tasks, differing in terms of the hypothetical vaccine’s efficacy (50% to 100%), the risk of serious side-effects (1 in 10,000 vs. 1 in 100,000), the location of the producer (the EU vs. the USA vs. China), and place of administration (general practitioner vs. local pharmacy vs. mass vaccination center). All of these dimensions were found to have some effect, yielding a difference in vaccine acceptance of approximately 15 percentage points between the most favored treatment (100% efficacy, 1:100,000 side effects, vaccines from the EU) and the least favored condition (50% efficacy, 1:10,000 side effects, vaccines from China).

It should be emphasized that these two studies were conducted before the COVID-19 vaccines were actually available. One may suspect that some respondents could thus perceive the question about their vaccination intentions as speculative and premature; it is likely that many had given the vaccines very little thought and so their opinions were relatively malleable. This could be one reason for the positive effects of experimental manipulations. 

Most recently, Serra-Garcia and Szech [19] asked American MTurk workers to make hypothetical choices between COVID-19 vaccination and gift cards of different values (within-subject), manipulating the default option (between-subject). They found that, compared with the baseline of 70% with no incentives, modest incentives (USD 20) reduced the declared take-up rate by 4.5 p.p., whereas substantial incentives (up to USD 500) increased it by up to 13.6 p.p.

Kluver et al. [20] addressed a large nationally representative sample in Germany. They exposed each of their respondents to two consequent choices about vaccine acceptance, manipulating three dimensions: whether there are financial incentives to vaccinate; whether vaccines are available at the local doctor’s surgery or only vaccination centers; and whether those vaccinated can enjoy freedom of travel. They found that the combined effects of all three strategies can increase vaccination uptake by as much as 13 percentage points among the undecided. 

Our own study builds upon these strategies, but investigates a much larger set of persuasive messages. It also uses a between-subject design, making it less susceptible to social desirability bias and other types of spillovers (at a cost of requiring a large number of observations to account for individual heterogeneity). In any case, we believe that the topicality of the issue, the dynamism of the pandemic situation, and cultural factors potentially affecting the results call for many more studies of this kind.

## 3. Materials and Methods

### 3.1. Sampling and Stimuli

We conducted a randomized online study with an emulated representative sample of nearly six thousand adult Polish internet users (n=3117 in Wave 1, n=2814 in Wave 2). The sample comes from the nationwide 110,000-strong survey panel Ariadna. Ariadna is a member of the European Society for Opinion and Marketing Research (ESOMAR) and has vast experience in running rigorous scientific surveys for many Polish academic institutions. The sociodemographic profile of people registered in Ariadna matches that of Polish internet users, and thus so does the sample in our study.

Ariadna only uses its own actively managed panel, which is built on the basis of multi-source recruitment. Recruitment is conducted through banners placed on websites and through mailings redirecting to the registration questionnaire, which ensures high ecological validity of the panel. In addition, groups with the lowest internet penetration are recruited through telephone interviews (CATI), direct interviews (CAPI, PAPI), and directly from personal databases. The sampling process is complemented by quota sampling according to key parameters to ensure representativeness and, if necessary, by sample weighting.

Invitations to participate in the study were sent only via email to the email addresses indicated by the participants. They received a message containing basic information about the survey along with a coded and personalized link. The survey link is valid for 48 h. After this time, a reminder is sent to those who have not completed the survey, as well as to those who have started the survey but have not completed it.

Each time a sample is selected for ongoing research, verification of some of the data from registration and start-up surveys is conducted. Telephone verification of randomly selected panelists is also applied, as well as the analysis of the consistency and coherence of responses in research questionnaires. Each panel member’s identity is verified, with personal data being confidential and responses to individual surveys being anonymous. Ariadna’s security measures exclude the activity of bots or any other virtual subjects. For each survey they fill in, the respondents earn virtual points that can be later spent in an online shop.

In Wave 1, the questions were appended to a longer questionnaire developed for another project. In that project, we asked a number of questions about COVID-19 (but not specifically about vaccines), unemployment, or the common cold (between-subject random assignment). Wave 2 was a stand-alone study, with some further changes and additions subsequently explored (see Appendix A for the exact wording of both questionnaires).

In both waves, prior to being asked about their willingness to get vaccinated, the respondents were exposed to a randomized information package presented in Table 1. In Wave 1, exactly three out of seven messages unrelated to the price were always displayed (and always followed the same order as in Table 1). This ensures the total length remains roughly constant but only allows us to compare the efficacy of different types of messages, not to ascertain if they are effective compared with no message at all. In Wave 2, for each potential message, there was an independent 50% chance that it is actually displayed (full factorial design). For example, only a randomly selected half of the subjects were told that the vaccine was developed by scientists from an international research consortium. 

In both waves, one of the four prices was always shown (multi-arm design). The price message was always placed at the end because three out of its four variants were, by necessity, counterfactual. We were afraid that whatever message following it could also be considered counterfactual. Other than that, the order of the messages was random. 

The two waves used slightly different sets of messages, as indicated in Table 1. Specifically, the vaccines available in Poland during Wave 1 were only those produced by Pfizer/Biontech. By contrast, Moderna and AstraZeneca vaccines were widely used by the time Wave 2 started. We thus opted for a more general formulation of the statement pertaining to producer reputation. Moreover, the message emphasizing the scarcity of vaccines was dropped because it was rather apparent that the vaccines were scarce. Instead, a message directly addressing common concerns that the vaccines are insufficiently tested was introduced.

The respondents were then asked if, provided the information they just read was confirmed, they would be willing to get vaccinated. They could choose between “definitely not”, “probably not”, “probably yes”, and “definitely yes”. To discourage mindless clicking, the respondents were not allowed to choose an answer in less than 10 s, but in practice, the median time spent on this question was much longer, namely 33 s (which is quite enough to read a few short sentences). In two follow-up open questions, they were asked to justify their response and indicate who or what could change their opinion [details of the method and procedures of the categorization of open-ended questions are presented in Appendix A)]. Those who said “definitely not” or “probably not” to the vaccination question were also asked whether they might change their opinion if they saw that the vaccine was effective and safe after the first few months of vaccinations. Additionally, they were asked about their attitude toward conspiracy theories propounded by anti-vaccinationists and pandemic non-believers. 

On top of standard demographic features and questions pertaining to emotions and risk attitudes, we elicited political orientation, feeling of control over the situation, feeling of being informed, and feeling worried (the latter three concerning the COVID-19 pandemic), religiosity, health conditions, mask-wearing, physical distancing, whether they or someone they know had COVID-19 and, if so, was hospitalized due to it. We also elicited predictions of the total number of confirmed COVID-19 cases and deaths due to COVID-19 during the next 12 months. Additionally, in the second wave, we asked about various trust levels (in the government, their neighbors, doctors, media, family, scientists) and about smoking. 

There was no difference in the central tendency of demographic variables between Waves 1 and 2 that would be significant at the 5% significance level, except for a slight difference in age (mean 43.7 years, median 44 in Wave 1; mean 45.8 years, median 46 in Wave 2), see Appendix A. 

### 3.2. Data Analysis Methods

We conducted both ordered logistic regressions on the original dependent variable with four levels (“definitely yes”, “probably yes”, “probably not”, and “definitely not” willing to be vaccinated) and logistic regressions with a binarized variable *vaxx_yes* (taking a value of 1 for those saying definitely or probably yes and zero otherwise).

We run a number of specifications with pre-registered sets of explanatory variables (see https://osf.io/e9cb2 accessed on 24 September 2021; a slight departure is that we skipped age squared, which is insignificant, to facilitate the interpretation and visualization of Figure 1. This has no bearing on other results in all models except for [1]. We also included 15 regional dummies that are jointly significant and that we overlooked in the pre-registered data analysis plan. Finally, we added per capita COVID-19 cases and deaths for the region and for the country as a whole from the preceding day (announced in the morning). They are not mentioned in the data analysis plan because it was not clear if they would be available. None of these changes have any qualitative bearing on the results). In model [1], we included our experimental variables and basic demographic variables. In [2], we additionally controlled for political preferences (the party the responder would vote for should elections be held next Sunday), self-declared emotions, risk preferences, and the extent to which the responder worries and feels informed about COVID-19. In further specifications, we built upon [2], seeking to identify possible moderation effects, in [4] we additionally allowed for interactions between experimental variables and key demographic variables; in [5], the interactions among experimental variables; in [6], those between self-reported material well-being and experimentally manipulated vaccination price; finally, in [7] interactions between experimental treatments and belief in conspiracy theories. 

We also ran some unregistered specifications suggested to us by early readers. These include interactions between gender and age, as well as between political preference and education level [3], and interactions between selected experimental treatments (statement about producer reputation (*v_producer_reputation)* and vaccine safety (*v_safety*)) and political preference [8]. We reported tables with odds ratios calculated for specifications 1–4 and 5–8 in Appendix A. The logistic regression in specification [3] also gives rise to Figure 1 and Figure 2.

Because of the previously described randomization procedure of Wave 1, we can only investigate the effect of any experimental treatment compared with a reference treatment. We chose the seemingly most subtle message, namely the one pertaining to scarcity (*v_scarcity*), as our baseline so that estimates for other treatments should be understood as additional effects compared with that one. 

## 4. Results

### 4.1. Wave 1

We first report the distribution of our main variable—willingness to get vaccinated (*v_decision*). Unless otherwise stated, throughout the paper, we report and analyze our data using post-stratification weights to better align the distribution of key demographic variables with those of the general population. However, this has no qualitative effect on the results. As shown in Table 2, our 3117 respondents are very split on the issue, with similar percentages choosing each of the four responses. 

Overall, our messages are ineffective; the specific estimates can be found in Appendix A. The only manipulation that makes a difference is that compared with the baseline of vaccines being available for free, the respondents would be even less willing to get vaccinated if asked to pay a modest amount of 70 PLN. 

We find strong demographic effects: being male, older, wealthier, and better educated (as well as worrying about COVID-19) makes respondents more likely to accept the vaccines. Those believing in conspiracy theories and those supporting right or ultra-right parties (or not voting at all), as well as those with the most intense religious practice, tend to be more negative. The interaction terms in models 4–8 are jointly insignificant. 

These key demographic effects are visualized in Figure 1 and Figure 2. Figure 1 shows that while gender and age are highly predictive, there is no interaction between them and no non-linear effect of the latter— the probability that a responder is willing to get vaccinated increases by about ¼ of a pct. point with each year of age and is about 7 pct. points higher in males than in females. Figure 2 confirms that voters on the extreme right are highly likely to be anti-vaxxers (especially those with only basic education), whereas other groups are more similar.

### 4.2. Wave 2

In Wave 2, a sample of 2814 respondents who had not taken part in Wave 1 was approached. In Table 3, we display the distribution of the dependent variable, which is very similar to that of Wave 1 (z=−1.558, p=0.1193 in a two-sample Wilcoxon rank-sum test run on unweighted data).

The analysis of data from Wave 2 is also analogous to that of Wave 1, with the exception that we do not need to treat any message as a baseline and that we incorporate additional explanatory variables available (notably trust towards various social groups). By contrast, given that Wave 2 was a stand-alone study, this time we do not have variables related to the study preceding Wave 1. We are also able to add model [9] testing for order effects. This includes interactions between messages and their position among all messages shown to the given responder and interactions between messages and dummies for them being shown as the very first message. Consequently, we can test if messages are more effective when shown early on (so that they are harder to miss or ignore). 

Again, we see that all messages are equally effective; this time, we can confirm our suspicion that all of them are indeed as effective as no message at all (Appendix A). In other words, they are ineffective. In none of the specifications is the effect of any of them on willingness to be vaccinated significant at the 5% level, thus rejecting the hypothesis that messages prove effective. Again, a price of PLN 70 strongly reduces respondents’ willingness to get vaccinated.

The demographic effects also match those reported in Wave 1, with males, older people, those with higher education, and greater wealth tending to be more pro-vaxx, although these variables are not significant in all the models. Again, political preferences and beliefs tend to be correlated with vaccination attitudes in predictable ways. For example, not voting or voting for the ultra-right (but this time not so much for the right-wing ruling party) is associated with a greater chance of opposing vaccines. Concerning the new variables, those trusting science, doctors, and the EU are more positive, as predicted. Respondents who believed to be in the risk group and who have had friends who have had COVID-19 are more willing to be vaccinated, while those believed to be allergic to vaccines less so. Thus, our pre-registered hypotheses concerning demographic effects are generally confirmed; the exception is that the positive effect of supporting the political left is not significant. The effects of health are weak and usually not significant. 

By contrast, hypotheses concerning moderation effects are soundly rejected, with interaction terms of models 4–7 being jointly insignificant (the only exception is that an unexpected *positive* interaction between conspiracy score and a statement about producer reputation (*v_prod_reputation*) leads to conspiracy–experimental treatment interactions of model [7] being significant). The interactions of models [8,9] are likewise insignificant (for the detailed regression output, see Appendix A). Overall, the results are very close to those of Wave 1. 

Figure 3 and Figure 4 are analogous to 1 and 2, respectively, again showing no interaction between sex and age and very little between education and political preference. Quantitatively, the effects of age and gender are very close to those identified in Wave 1. As for political preference, we see a more pro-vaccine attitude among the supporters of the right-wing ruling party compared with Wave 1. One explanation for this could be that early on, the vaccines were primarily associated with the pharma companies that have produced them and the European Commission orchestrating their purchase; over time, they have become more associated with the government organizing their distribution. This might have made them more acceptable among the voters of the ruling party, explaining the slight shift in the attitude between the waves.

## 5. Discussion and Conclusions 

In this paper, we report the results of two waves of a survey using a large and diverse sample. It thus allows the investigation of even subtle effects, such as the evolution of attitudes towards vaccines among supporters of a specific party over time. We also observe a number of strong main effects, largely confirming findings on the determinants of attitudes towards COVID-19 vaccines reported previously for other countries. Furthermore, the general level of vaccine hesitancy is comparable to those reported previously in smaller samples in Poland as in Sowa [16]. 

Still, the main experimental finding is negative—we were not able to persuade the respondents to change their opinion. One reason for that could be that respondents did not even bother to read our messages. This could be true for some of them, and these people may also be prone to ignoring persuasive messages sent via TV, radio, billboards, etc. In this sense, even if our null result is due to inattention, it may well have external validity. More importantly, we have clear evidence that a substantial fraction of our respondents did read the messages carefully. First, median reading times did not suggest mindless clicking, as mentioned previously. Second, our respondents clearly did *not* miss the message about the vaccines costing 70 PLN—it made them substantially less likely to respond positively. 

Third, more evidence comes from the responses to the open-ended questions. We observe that our manipulations could not have been missed altogether because they did affect these responses, as seen in the prevalence of some categories. Most spectacularly (and expectedly), the fraction of respondents who complain about the vaccines being too costly is 6.6% in the condition where the hypothetical vaccine price was PLN 70 compared with just 1.0% otherwise. More than that, there are also significant differences for manipulations that do not affect declared intentions. For example, the prevalence of mentioning convenience as a reason to get vaccinated (*why_convenience*) is 9.6% with the statement pertaining to the vaccine passport (*v_vax_passport*) manipulation and just 6.1% without, a significant difference.

Interestingly, some of these differences suggest another interpretation of the null result—in some respondents, the manipulations could have backfired. For example, 8.0% of respondents mentioned their concern that the vaccines might have been poorly tested (*why_poor*) when we explicitly addressed this issue with the statement that vaccines have been thoroughly tested (*v_tested* equal to one) compared with only 6.3% otherwise, p=0.039 in a one-sided test of proportions. In other words, our manipulation actually made some respondents consider the possibility that the vaccines may be insufficiently tested, most probably making their attitude more negative. 

Likewise, given that PLN 70 is better than nothing, our observation that vaccine acceptance was identical in the free condition and in the ‘patient pays 70 PLN’ condition suggests that the latter made the vaccine per se look less attractive in some respondents’ eyes. On top of the crowding-out of intrinsic motivation as suggested i.a. by Serra-Garcia and Szech (19) (a warm glow is lost if somebody is paid for their good deed), this condition could trigger suspiciousness (“there must be something wrong with it if they pay me to take it”). 

There are other plausible reasons for the null result. People might have just heard too much about the virus and the vaccines over the last year; they might ignore or even avoid information [21]. Understandably, the virus and the pandemic it caused may evoke strong emotional reactions. It is subsequently safe to assume that most people rely primarily on affective/experiential rather than analytic/rational systems [22] to assess risks and make decisions [23]; this means that altering these decisions using rational arguments and statistics is very difficult. Instead, campaigns could focus on changing the emotions associated with vaccinations. 

Given the design of our study, we did not include any audio or visuals, instead focusing on plain text messages. Voluminous marketing literature testifies to the importance of visuals; then again, in the context of vaccination hesitancy it is mostly their importance in conveying “complex risk information” [24] that is emphasized. There was no such information in our stimuli, suggesting that the lack of visuals may have been less of an issue. Still, it is possible that an image of a celebrity, a physician, or a nurse could help build more positive associations of the COVID-19 vaccines. Vaccination promotion by religious leaders could be effective too, given the likely affective/experiential nature of vaccination decisions and significance of the religiosity dimension in our models. 

Finally, low levels of trust in Poland, particularly a low level of trust towards public institutions [25,26], makes effective public campaigning a challenge indeed. For example, in our sample, as many as 70% said “no” (rather than “yes, to a large extent”, “yes, somewhat”, or “no opinion”) when asked if they trusted the government. 

In either case, our results suggest that information campaigns may be misguided. This finding backs up the conjectures made by several experts; “It’s a reasonable thing not to have some giant national campaign,” as UCLA professor Hal Hershfield told USA Today [27].

We propose three alternative lines of action. First, persuasion could be tailored to individual reasons for vaccine hesitancy, identified using social media or personal interviews. For example, 3.5% of our respondents declining vaccines (usually choosing “probably not” instead of “definitely not”) justified it in terms of individual contraindications, most typically allergies and asthma. These fears are overblown. Specifically, according to WHO, the prevalence of severe allergic reactions to COVID-19 vaccines is approximately one in a million, and asthma is not a contraindication. It seems very plausible that a short conversation with a trusted physician could change these people’s opinions. A study of vaccination uptake in elderly Poles indicates that this approach could indeed be effective [28]. Likewise, in Turkey, a campaign involving face-to-face contact with doctors and community leaders boosted the vaccine take-up rate by nearly 30% [29]. In any case, our results can be directly used to predict any individual’s COVID-19 vaccine attitude, so that any persuasive attempts are focused on those that are relatively likely to change their mind rather than die-hard anti-vaxxers for whom any such attempt would be futile. 

Second, it seems more promising to work indirectly, trying to turn those willing to be vaccinated into champions of the campaign. To some extent, this is already happening, with millions of people adding “I got my COVID-19 vaccine” or a similar frame to their Facebook profile picture. There is, however, no reason not to hand out physical stickers to those just vaccinated, for example, so that they can advertise beyond their social media bubbles. They could also be encouraged to talk about their experience and the side effects included with their friends and family members. As most people tend to infer too much from small samples, the fact that, say, five friends were vaccinated and none of them had any serious symptoms could be more convincing than statistics covering millions of cases. 

Third, more attention should perhaps be devoted to interventions that are likely to work, such as making sure that those who are in principle willing to be vaccinated actually follow through. One way to achieve that involves the use of reminders (see [30] for a review of earlier literature and [31] for a recent case involving COVID-19 vaccines). Another simple intervention that has been shown to work [32] involves prompting individual patients to write down the date and time of the appointment. To the extent that these measures may prove insufficient to assure collective immunity, a serious discussion of various forms of coercion should likely be commenced.

All of these, however, are partial, short-term solutions. A more fundamental change is needed, in Poland and elsewhere, in the education system to effectively teach critical thinking and the ability to assess the trustworthiness of different sources of information in the complex digital world. Without it, we remain highly vulnerable to global threats requiring new types of large-scale voluntary collaboration based on scientific insights.

## Figures and Tables

**Figure 1 vaccines-09-01113-f001:**
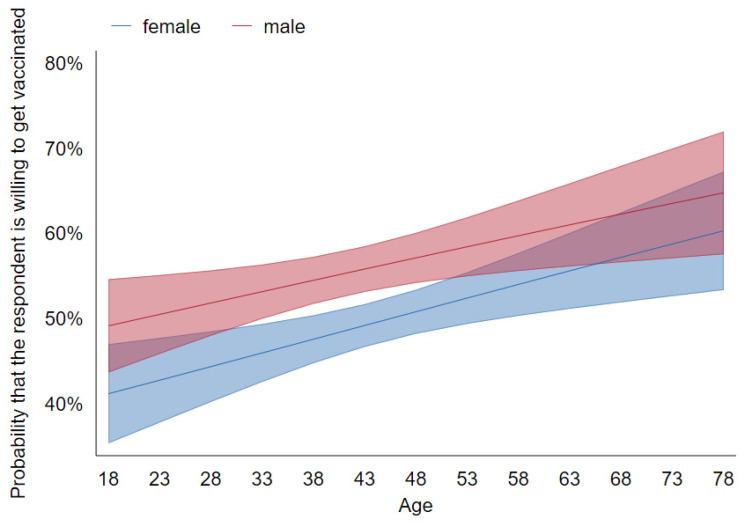
Predictive margins of sex and gender on vaccination decision with 95% confidence intervals (CIs; Wave 1).

**Figure 2 vaccines-09-01113-f002:**
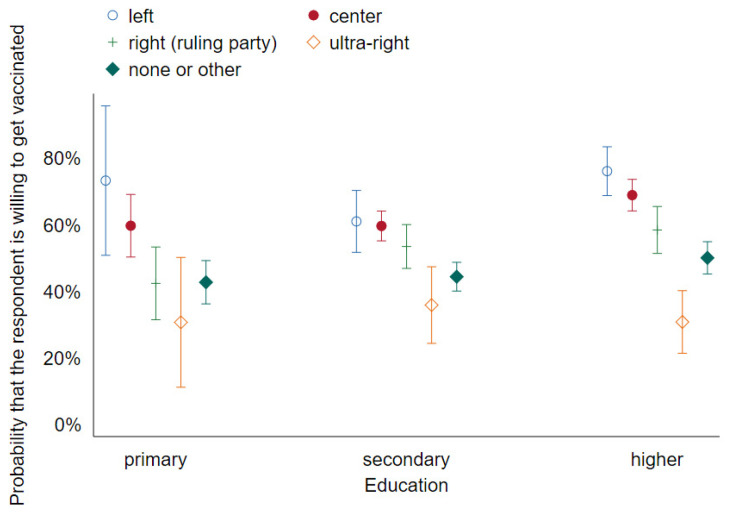
Predictive margins of education and voting preferences on vaccination decision with 95% CIs (Wave 1).

**Figure 3 vaccines-09-01113-f003:**
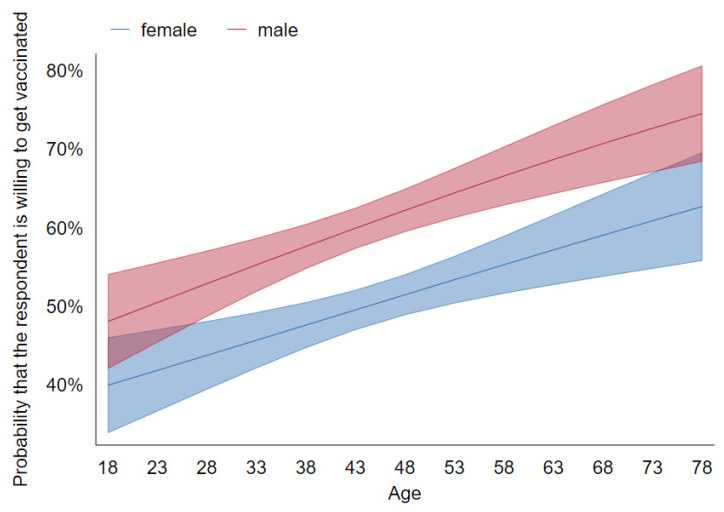
Predictive margins of sex and gender on vaccination decision with 95% CIs (Wave 2).

**Figure 4 vaccines-09-01113-f004:**
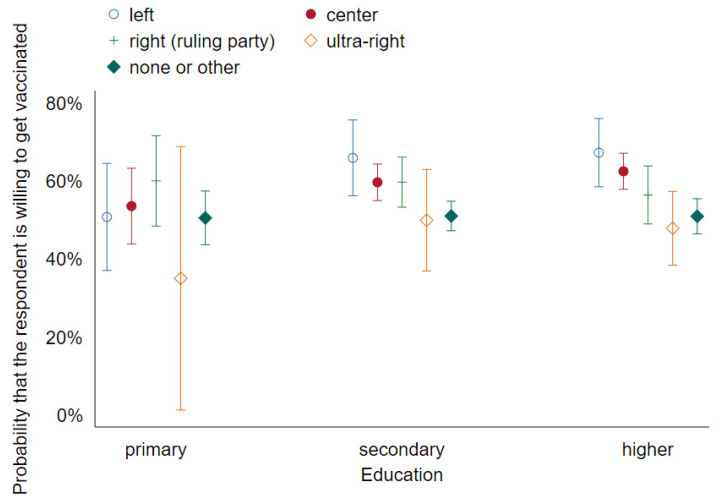
Predictive margins of education and voting preferences on vaccination decision with 95% CIs (Wave 2).

**Table 1 vaccines-09-01113-t001:** Messages employed in the study.

*Common Sentence for All Subjects:* A Vaccine for Coronavirus has Recently Become Available in Poland Vaccination is Voluntary.	
*Then, the following persuasive messages—each could be present or not.* *Wave 1: exactly three randomly selected messages were shown* *Wave 2: each message was independently drawn with a 50% chance*	
Producer reputation *(v_producer_reputation)*	The vaccine was developed by scientists from the American Pfizer and the German company Biontech. [Wave 1]/The vaccine was developed by scientists from an international research consortium. [Wave 2]
Efficiency *(v_efficiency)*	The vaccine’s effectiveness has been estimated at over 90%, which means that a vaccinated person is more than ten times less likely to get the disease than an unvaccinated person.
Safety *(v_safety)*	The European Medicines Agency confirms that the vaccine is safe. Possible side effects are mild to moderate, can be treated with paracetamol, and disappear within a few days.
Others want it *(v_other_want_it)*	Research conducted by IPSOS on 18,000 people in 15 countries shows that about 75% want to get vaccinated as soon as possible.
Scientific authority *(v_scientific_authority)*	According to the COVID-19 team at the Polish Academy of Sciences, “vaccination is the only rational choice, thanks to which we will be able to exit the pandemic faster.” The use of the vaccine is also recommended by the Supreme Medical Chamber and many other medical and scientific societies.
Vaccine passport *(v_vax_passport)*	It should be assumed that vaccination will make everyday life easier: vaccinated people will not have to quarantine after contact with an infected person, will be able to travel freely abroad, will not have to wear a face mask, etc.
Scarcity * *(v_scarcity)*	In the initial stages, there will not be enough vaccines for everyone.
Thoroughly tested ** *(v_tested)*	Development work on the vaccines began immediately after the pandemic outbreak and was treated as a priority. It drew on the vast experience of the research teams involved and used some of the solutions that had been used in vaccines for years. In total, more than 100,000 people were tested in clinical trials.
*Price information: one of four versions was randomly shown:*	
Patient pays 0 (free vaccine) *(v_p_pays0)*	Now suppose that the vaccine will be free for the person who wants to be vaccinated.
Patient gets 70PLN *(v_p_gets70)*	Now suppose that the vaccine will be free for a person who wants to be vaccinated, and as an incentive for mass vaccination, the government will pay everyone who wants to be vaccinated PLN 70 [ca. EUR 15].
Patient pays 10PLN *(v_p_pays10)*	Now suppose that the vaccine will be fee-based and will cost about PLN 10 per person.
Patient pays 70PLN *(v_p_pays70)*	Now suppose that the vaccine will be fee-based and will cost about PLN 70 per person.

*Source code variable name in italic.* * included in Wave 1 only. ** included in Wave 2 only.

**Table 2 vaccines-09-01113-t002:** Distribution of willingness to get vaccinated.

Vaccination Intention	Definitely Yes	Probably Yes	Probably Not	Definitely Not
Fraction	25.9%	26.6%	26.5%	21.1%

**Table 3 vaccines-09-01113-t003:** Distribution of willingness to get vaccinated.

Vaccination Intention	Definitely Yes	Probably Yes	Probably Not	Definitely Not
Fraction	24.1%	30.8%	24.6%	20.5%

## Data Availability

The data will be available upon reasonable request to the corresponding authors.

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
