# Peer review of "Persuasive Messages Will Not Increase COVID-19 Vaccine Acceptance: Evidence from a Nationwide Online Experiment"

_vaccines, 2021, doi:10.3390/vaccines9101113_

Round 1

Reviewer 1 Report

This is an interesting paper on an experiment to test the effectiveness of pro-vax messages for COVID-19. The paper requires some minor-to-moderate revisions before it may be considered for publication.

  1. There are some awkward sentences and a thorough copy edit by a native English speaker should be done.
  2. There is no description of the data analysis approach or software used. This should be added.
  3. There is very limited description of measures and the survey methodology. This should be expanded.
  4. When describing significant results in the results section (e.g., demographic effects), the actual statistics from the models shown in tables/figures should be described.
  5. So it appears from the text that waves 1 and 2 are cross-sectional and no statistical comparisons are made. While the data are not longitudinal, cross-sectional comparisons could be made. Why did the authors not do this? Provide some explanation and also clarify the analysis procedure (earlier comment).
  6. Another hypothesis, not discussed in the discussion section, is that the messages were presented as bland text with no graphics, color, framing, or other persuasive marketing elements. There is published literature on the use of elaborated marketing strategies to reduce vaccine hesitancy and promote vaccination. The authors should at least acknowledge this as a potential alternative explanation of their negative findings.

Author Response

This is an interesting paper on an experiment to test the effectiveness of pro-vax messages for COVID-19. The paper requires some minor-to-moderate revisions before it may be considered for publication.

  • Thanks for your kind assessment and constructive remarks.

1.      There are some awkward sentences and a thorough copy edit by a native English speaker should be done.

  • We have now employed a professional editor, a native speaker with relevant academic background, to proofread the text.

2.      There is no description of the data analysis approach or software used. This should be added.

  • As indicated in the manuscript, all the details are provided in the preregistered data analysis plan. We have now added a sub-section on the data analysis methods and information about the software used in the manuscript; thank you for spotting this omission.

3.      There is very limited description of measures and the survey methodology. This should be expanded.

  • We have now provided more detailed information on the Ariadna panel and other aspects of the survey methodology (e.g., how non-responses were handled).

4.      When describing significant results in the results section (e.g., demographic effects), the actual statistics from the models shown in tables/figures should be described.

  • We have added some numbers for the key demographic effects (gender and age). We have also visualized the odds ratios using figures (in appendices).

5.      So it appears from the text that waves 1 and 2 are cross-sectional and no statistical comparisons are made. While the data are not longitudinal, cross-sectional comparisons could be made. Why did the authors not do this? Provide some explanation and also clarify the analysis procedure (earlier comment).

  • We have now tested for differences in the distribution of demographic variables in the two waves.

6.      Another hypothesis, not discussed in the discussion section, is that the messages were presented as bland text with no graphics, color, framing, or other persuasive marketing elements. There is published literature on the use of elaborated marketing strategies to reduce vaccine hesitancy and promote vaccination. The authors should at least acknowledge this as a potential alternative explanation of their negative findings.

  • We completely agree. We have now addressed this issue in the discussion section.

Reviewer 2 Report

Estimated Authors of the paper: "Persuasive messages will not raise COVID-19 vaccine acceptance. Evidence from a nation-wide online experiment",

I've read with great interest your research paper reporting on effectiveness of pro-vaccination messages in an online experiment conducted in February-March 2021 with a sample of almost six thousand adult Poles. In summary, your estimates suggest that nearly 45% of the responders were unwilling to be vaccinated and none of the popular messages we used was effective in reducing this hesitancy.

This is quite interesting, and may be of certain interest for healthcare professionals participating into the next stages of mass vaccination campaigns, not only in Poland but also in Europe.

Unfortunately, in my opinion, the present paper must be extensively reworked before its final acceptance. In fact, there are some significant shortcomings, and more precisely:

1) the text must be amended avoiding the referral to the source code designation of variables; even though such attitude has a certain rationale, it also impairs the proper understanding of the main text. Please report the variables in a plain way.

2) please provide some further information about Ariadna; how the participants were originally recruited and eventually address possible sampling bias and subsequent impairment to the overall representitivity of the study sample in confront of the original population.

3) please include a summary of demographic characteristics of the study participants, and also please compare the two subpopulation (wave 1 vs. wave 2) in order to confirm or dismiss potential differences in the sampled participants

4) the text too often asks the reader to check the supplementary materials; if a table or a figure is required to follow the text and its meaning, it should be embedded in it, not marginalized to annex materials. Please rework the organization of your tables accordingly.

some minor but still significant remarks:

Row 243: please either avoid party names (the actual settings may be scarcely appreciable for people unaware of the Polish political situation) by opting in for descriptive reporting (e.g. far right, right, nationalists, etc., obviously if it is possible).

row 182 following: this section should be moved to the methods section

row 206 following: the narrative reporting of the data is ok; please include also the discrete details as a Table of forrest plot alike Figure 2

Figure 2/4: please rework the figure in order to include both captions and labels of the axes consistent with the main text

After the aforementioned amendments, I think that the present paper may be checked again, and possibly reconsidered for a full publication on Vaccines.

Author Response

I've read with great interest your research paper reporting on effectiveness of pro-vaccination messages in an online experiment conducted in February-March 2021 with a sample of almost six thousand adult Poles. In summary, your estimates suggest that nearly 45% of the responders were unwilling to be vaccinated and none of the popular messages we used was effective in reducing this hesitancy.

This is quite interesting, and may be of certain interest for healthcare professionals participating into the next stages of mass vaccination campaigns, not only in Poland but also in Europe.

Unfortunately, in my opinion, the present paper must be extensively reworked before its final acceptance. In fact, there are some significant shortcomings, and more precisely:

1) the text must be amended avoiding the referral to the source code designation of variables; even though such attitude has a certain rationale, it also impairs the proper understanding of the main text. Please report the variables in a plain way.

  • We have now changed the variable names to plain language; thanks for this helpful suggestion. We have kept the variable names (in parenthesis and italics) to facilitate their identification in the source code or coefficient tables.

2) please provide some further information about Ariadna; how the participants were originally recruited and eventually address possible sampling bias and subsequent impairment to the overall representitivity of the study sample in confront of the original population.

  • We have now provided more detailed information on the Ariadna panel, the recruitment process, and some other aspects of the survey methodology.

3) please include a summary of demographic characteristics of the study participants, and also please compare the two subpopulation (wave 1 vs. wave 2) in order to confirm or dismiss potential differences in the sampled participants

  • We have provided summary statistics in Table C3.1 in Supporting Information, tested for differences in the distribution of demographic variables in the two waves, and added this information into the main text.

4) the text too often asks the reader to check the supplementary materials; if a table or a figure is required to follow the text and its meaning, it should be embedded in it, not marginalized to annex materials. Please rework the organization of your tables accordingly.

  • Thanks for this comment. We have tried to accommodate it in its entirety but this would mean cluttering the text with an additional 10 pages of reader-unfriendly material (Tables C1.1-4 and C2.1-4). We chose not to do this, instead just reporting the main findings in the text. However, if you insist that it is necessary to include all these tables (and the Editor accepts it), we have no problem doing so.

  • Additionally, by adding the “Data analysis methods” section we contain the number of mentions of the appendix; we have also rephrased numerous sentences throughout the text meaning that viewing the appendix is optional but not crucial for readers’ comprehension.

some minor but still significant remarks:

Row 243: please either avoid party names (the actual settings may be scarcely appreciable for people unaware of the Polish political situation) by opting in for descriptive reporting (e.g. far right, right, nationalists, etc., obviously if it is possible).

  • We have removed the party names.

row 182 following: this section should be moved to the methods section

  • This has now been done. More specifically, we have also moved the paragraph ending in line 182 to the methods section.

row 206 following: the narrative reporting of the data is ok; please include also the discrete details as a Table of forrest plot alike Figure 2

  • We have added forest plots to the Supporting Information file. Again, we would have no problem including the two of them in the main text.

Figure 2/4: please rework the figure in order to include both captions and labels of the axes consistent with the main text

  • Done

After the aforementioned amendments, I think that the present paper may be checked again, and possibly reconsidered for a full publication on Vaccines.

  • Again, thanks for your assessment and helpful suggestions.

Round 2

Reviewer 1 Report

The paper is much improved and should be published after a copyedit/proof.

Reviewer 2 Report

Estimated Authors,

I've greatly appreciated the considerable efforts you paid in order to improve the overall quality of this paper.

At the moment, I've only a very minor suggestion:

in the introduction, you referred to the term "social ad campaign". After the conventional English editing by MDPI publishing editors, please change as "social advertising campaign", by checking if this terminology is otherwise used across the text, and then change it accordingly - I give you this suggestion as not all Public Health professionals are familiar with this terminology.

Again: well done!